# Time-varying SUVr reflects the dynamics of dopamine increases during methylphenidate challenges in humans

Dardo Tomasi [1✉], Peter Manza [1], Jean Logan[2], Ehsan Shokri-Kojori [1], Michele-Vera Yonga[1], Danielle Kroll [1], Dana Feldman[1], Katherine McPherson [1], Catherine Biesecker[1], Evan Dennis[1], Allison Johnson[1], Kai Yuan[3], Wen-Tung Wang[4], John A. Butman [4], Gene-Jack Wang [1] & Nora D. Volkow [1]

Dopamine facilitates cognition and is implicated in reward processing. Methylphenidate, a dopamine transporter blocker widely used to treat attention-deficit/hyperactivity disorder, can have rewarding and addictive effects if injected. Since methylphenidate's brain uptake is much faster after intravenous than oral intake, we hypothesize that the speed of dopamine increases in the striatum in addition to its amplitude underly drug reward. To test this we use simulations and PET data of [$^{11}$C]raclopride's binding displacement with oral and intravenous methylphenidate challenges in 20 healthy controls. Simulations suggest that the time-varying difference in standardized uptake value ratios for [$^{11}$C]raclopride between placebo and methylphenidate conditions is a proxy for the time-varying dopamine increases induced by methylphenidate. Here we show that the dopamine increase induced by intravenous methylphenidate (0.25 mg/kg) in the striatum is significantly faster than that by oral methylphenidate (60 mg), and its time-to-peak is strongly associated with the intensity of the self-report of "high". We show for the first time that the "high" is associated with the fast dopamine increases induced by methylphenidate.

[1] National Institute on Alcohol Abuse and Alcoholism, National Institutes of Health, Bethesda, MD, USA. [2] Center for Biomedical Imaging, Department of Radiology, New York University School of Medicine, NY New York, USA. [3] School of Life Science and Technology, Xidian University, Xi'an, Shaanxi, China. [4] Radiology and Imaging Sciences, Clinical Center, National Institute of Health, Bethesda, MD, USA. ✉email: dardo.tomasi@nih.gov

Dynamic measures of dopamine (DA) are needed to investigate the rate at which stimulants or other addictive drugs change DA signaling in the brain. The rewarding effects of addictive drugs are much stronger when they are injected, presumably due to their faster brain delivery compared to when they are taken orally, which results in much slower brain uptake[1]. Here, we show that a simple approach using positron emission tomography (PET) can be used to non-invasively assess the dynamics of extracellular dopamine increases induced by methylphenidate (MP) in the human brain, when given intravenously versus when given orally.

Like cocaine, MP blocks DA transporters (DAT)[2], thus inhibiting DA reuptake and increasing extracellular DA[3,4]. When MP is misused for its rewarding effects it is predominantly snorted or injected[5], which results in much faster brain delivery than when it is taken orally as is the case when used clinically for the treatment of attention deficit hyperactivity disorder (ADHD)[6]. These different behavioral effects suggest that the rate of DA signaling in brain reward regions is a crucial variable for drug reinforcement[7]. However, while this association has been inferred[2,7] it has not been directly confirmed.

Several models have been proposed to assess the effects of time-varying concentrations of endogenous dopamine on the binding of $D_{2/3}$ receptor ligands such as [$^{11}$C]raclopride[8-17]. The linear extension of the simplified reference region model (LSSRM), which models the dynamics of dopamine-radioligand competition binding in terms of time-varying changes in efflux rate, $\gamma h(t) = \exp(-\lambda t)$, following stimulation at $t = 0$ is a popular method to predict single-scan PET signal changes induced by task-related changes in endogenous dopamine levels[18-23] and pharmacological challenges[24,25]. The neurotransmitter PET[26] (ntPET), a dual-scan approach for assessing increases in dopamine concentration induced by a pharmacological challenge (scan 2) compared to baseline (scan 1), was originally demonstrated in rats using alcohol[26] and methamphetamine[27] challenges. The linearization of the parametric ntPET approach using gamma variate functions for the time course of endogenous DA increases was proposed for the LSSRM to model non-instantaneous DA increases[15], and used in humans to estimate the dynamics of DA increases in the striatum during smoking[28,29].

Here, we carried out a series of simulations to test the hypothesis that the time-varying SUVr-difference between [$^{11}$C]raclopride scans collected with and without a pharmacological MP-challenge reflects the dynamics of dopamine increases induced by MP in the striatum. To test the hypothesis that the intensity of the 'high' reflects the rate of dopamine increases in the striatum, we carried out a within-subject [$^{11}$C]raclopride PET study with a double-blind placebo-controlled design in twenty healthy adults. We studied dynamic dopamine increases using oral- (slow drug brain delivery) and intravenous (IV)-MP (fast drug brain delivery) challenges, in association with measured subjective responses to MP using self-reports of 'high' throughout the scan.

## Results

**ΔSUVr simulations.** We first examined if the dynamics of DAT occupancy or the dynamics of DA increases predict the time-varying changes in standardized uptake value ratios to cerebellum (SUVr) elicited by MP in simulated PET data. Dynamic SUVr-changes (ΔSUVr) between placebo and MP conditions were simulated using Eqns [2] and [3] (see Methods), and differed across two alternative mechanistic models for the PET signal; one assuming that $h(t)$ is proportional to the fractional occupancy of DAT by MP, $f_{occ}^{DAT}(t)$, and the other that $h(t)$ is proportional to DA increases induced by MP, $D(t)$ (see Methods) (Fig. 1).

Specifically, we found that the slower increase of $h(t)$ after IV-MP for $D(t)$ than for $f_{occ}^{DAT}(t)$ (Fig. 1a) translated into a 10 min delay between the simulated ΔSUVr curves (Fig. 1b). Qualitatively, ΔSUVr dynamics were similar to that of $h(t)$ when $h(t) \propto D(t)$ (Fig. 1d) but not when $h(t) \propto f_{occ}^{DAT}(t)$ (Fig. 1c). These simulations suggest that the dynamics of endogenous dopamine increases, but not that of DAT occupancy, shape the time-varying ΔSUVr elicited by MP. The simulations also showed that the amplitude of ΔSUVr was sensitive to MP dose, such that increasing MP doses resulted in sublinear ΔSUVr increases (Fig. 2). Note that the dynamics of ΔSUVr(t), assessed with simulations using a gamma variate function for $h(t)$[15], which has been successfully used to model preclinical dynamic PET data with a serotoninergic challenge[30], did not correspond with the dynamics of the experimental ΔSUVr(t) data with MP in the present study.

Because $D(t)$ relied on pharmacokinetic parameters from the literature[31] that may not represent well the pharmacokinetic characteristics of the participants in the present study, we tested whether a gamma cumulative distribution function (CDF), $F(t)$, fitted to the data using Eqn [8] could be used as a proxy for $D(t)$. Since $D(t)$ and $F(t)$ had similar temporal profiles and fitted the simulated ΔSUVr(t) data similarly (Fig. 3a), and their time derivatives, $d(t)$ and $f(t)$ (a gamma probability density function), had similar time-to-peak (TTP) (Fig. 3b), we interpreted ΔSUVr(t) as dynamic DA increases induced by MP, and used $f(t)$ to measure dopamine increase TTP.

Since ΔSUVr(t) is based on dual-scan experiments[10,26] that rely on point-by-point subtractions and may require two identical scan protocols differing solely by the presence/absence of MP we investigated the effect of variability on protocol execution as well as within-experiment physiological variability in 1000 simulations. Specifically, 120 s (SD) random variations of injection times (raclopride or MP) caused minimal variability in fitted TTP (δTTP: 0.37 min, for the raclopride bolus and 0.67 min, for the IV-MP bolus; SD). Similarly, 4% variability (100 s.d./mean) in raclopride's input function parameters ($t_{peak}$, $A_i$ and $T_i$) and LSSRM pharmacokinetic parameters ($K_1$, $R_1$, $k_2$, and $k_{2a}$) between placebo and IV-MP, caused minimal variability in fitted TTP (δTTP: 0.63 min) suggesting that fitted TTP is not sensitive to precise injection times (1000 simulations). These simulations suggest that fitted TTP is not particularly sensitive to precise protocol execution (injection times) or within-subjects physiological variability.

To assess the sensitivity of TTP to MP dose we simulated the normal variability of ΔSUVr dynamics within and across individuals (1000 simulations). For IV-MP, random variations in LSSRM parameters (10%) also caused minimal variability in fitted TTP (mean δTTP < 1 min). Differently, random variations in MP dose, input function, and LSSRM parameters (10%) showed that fitted TTP was significantly associated with the variabilities of the MP dose ($R^2 = 0.6$) and the decay rate of MP blood concentrations ($\lambda$, $R^2 = 0.02$), but not with the variability of other parameters (Fig. 4). These simulations suggest that fitted TTP, which is highly sensitive to MP dose, is only modestly influenced by $\lambda$. For oral-MP, random variations in LSSRM parameters (10%) also caused minimal variability in fitted TTP (mean δTTP < 1 min), and random variations in MP dose, $t_{peak}$, and LSSRM parameters (10%) showed that fitted TTP was significantly associated with the variabilities of $t_{peak}$ of plasma MP ($R^2 = 0.61$) and MP dose ($R^2 = 0.1$), but not with the variability of other parameters (Fig. 5).

Similarly, we assessed the association between fitted TTP and the 'true' DA rate TTP with $\lambda = 0.05$ min$^{-1}$ and 4% random variability in all other parameters (1000 simulations) and found that fitted and 'true' DA rate TTP had excellent correlation

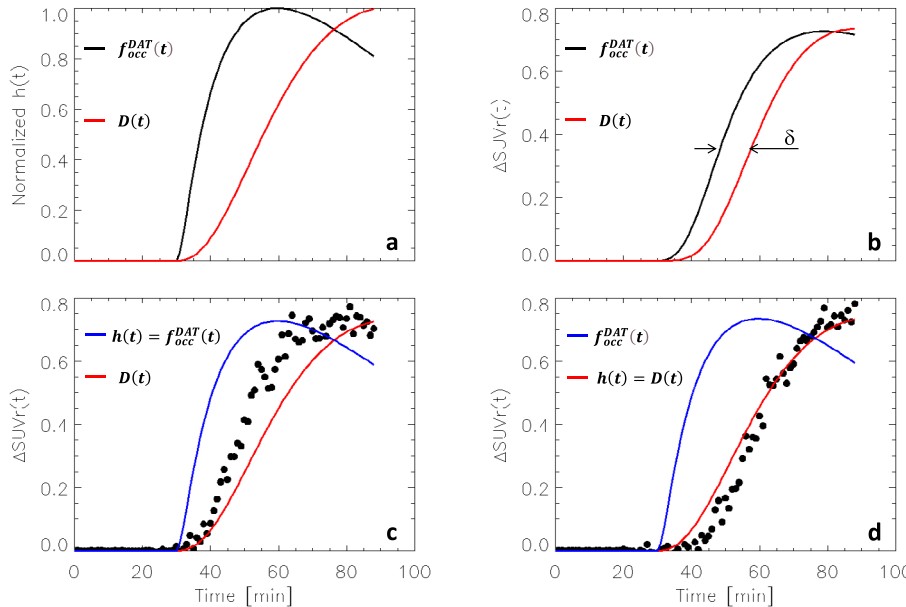

**Fig. 1 SUVr simulations: Specificity.** Time-varying fractional occupancy of dopamine transporter (DAT), $f_{occ}^{DAT}(t)$, and relative extracellular dopamine increases, $D(t)$, were simulated using Eqns [6] and [8] and used as alternative mechanistic models for $h(t)$, which is normalized to 1 and reflects the dynamics of binding competition between raclopride and dopamine increases induced by methylphenidate **a**. Noiseless time-varying differences in standardized uptake value ratios to the cerebellum (ΔSUVr) in striatum between placebo and intravenous methylphenidate (IV-MP) injected at t = 30 min, simulated using Eqns [2] and [3], showing the time delay (δ) between simulations with $h(t) \propto f_{occ}^{DAT}(t)$ or $D(t)$ **b**. The dynamics of the normalized $f_{occ}^{DAT}(t)$ did not resemble the dynamics of simulated ΔSUVr (dots) when $h(t) \propto f_{occ}^{DAT}(t)$ but that of $D(t)$ did it when $h(t) \propto D(t)$ **c**, **d**. Normal random noise (3%) was added to the SUVr time courses for placebo and IV-MP prior to compute ΔSUVr in **c** and **d**. $K_1^{MP} = 0.6$ min⁻¹, $k_2^{MP} = 0.06$ min⁻¹, $k_3^{MP} = 0.5$ min⁻¹, and $k_4^{MP} = 0.2$ min⁻¹ (see ref. [31]); $K_{1r} = 0.092$ mL/min g, $k_{2r} = 0.45$ min⁻¹, and $R_1 = 1.154$, $k_2 = 0.45$ min⁻¹, and $k_{2a} = 0.065$ min⁻¹ (see ref. [30]); β = 0.02 min⁻¹.

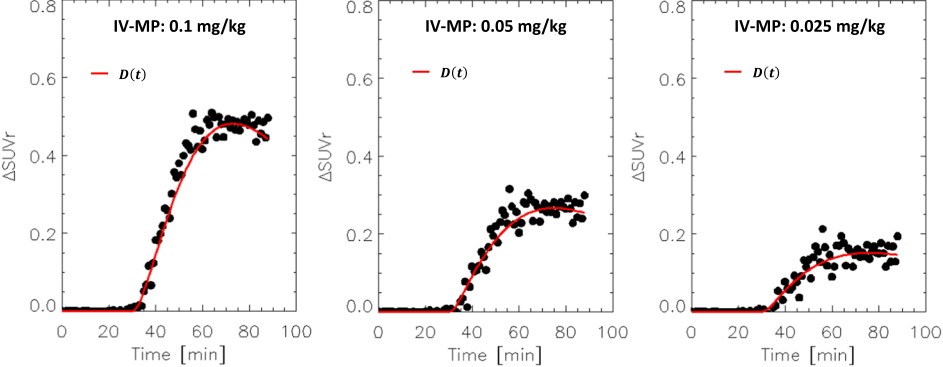

**Fig. 2 SUVr simulations: Sensitivity.** Dynamic dopamine increases, $D(t)$, simulated with Eqns [6] and [8], and the corresponding changes in standardized uptake value ratio (ΔSUVr) between placebo and intravenous methylphenidate (IV-MP) simulated with Eqns [2] and [3] for $h(t)=D(t)$ and 3 different pharmacological doses of MP. Normal random noise (3%) was added to the SUVr time courses for placebo and IV-MP prior to compute ΔSUVr. $K_1^{MP} = 0.6$ min⁻¹, $k_2^{MP} = 0.06$ min⁻¹, $k_3^{MP} = 0.5$ min⁻¹, and $k_4^{MP} = 0.2$ min⁻¹ (see ref. [31]); $K_{1r} = 0.092$ mL/min g, $k_{2r} = 0.45$ min⁻¹, and $R_1 = 1.154$, $k_2 = 0.45$ min⁻¹, and $k_{2a} = 0.065$ min⁻¹ (see ref. [30]); β = 0.02 min⁻¹.

(R = 0.76 or 0.61, IV- or oral-MP; Figs. 4b and 5b) and a mean difference | δTTP | = 5 ± 2 min (Figs. 4c and 5c). These simulations suggest that fitted TTP can predict DA rate TTP.

**ΔSUVr dynamics in humans.** Next, we assessed the dynamics of ΔSUVr in 20 healthy participants. As expected, [¹¹C]raclopride's binding was high in the striatum and low in other brain regions. To study the availability of D_{2/3} receptors in the striatum we mapped the non-displaceable binding potential (BPnd) using a graphical method that does not require blood sampling. BPnd was lower, both for IV- and oral-MP, than for placebo

demonstrating significant DA increases in the striatum for the static BPnd measures extracted from the 90 min scans (P_FWE < 0.05; Fig. 6a). However, the BPnd-difference between placebo and MP (ΔBPnd) was not significantly different for IV- than for oral-MP (P = 0.44, F(1,38) = 0.6, within-subjects ANOVA). These data indicate that conventional PET static methods for estimating DA increases do not have the necessary sensitivity for detecting differences in DA increases between oral and IV administration routes at the doses used in this study.

We estimated time-varying DA increases in putamen, caudate and ventral striatum by contrasting striatal SUVr time courses for placebo and MP conditions. The dynamic analysis based on

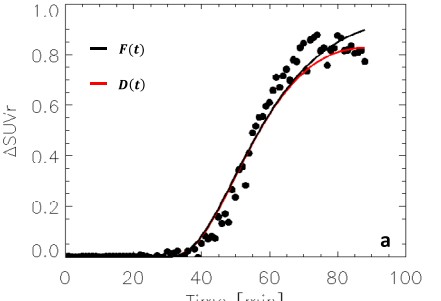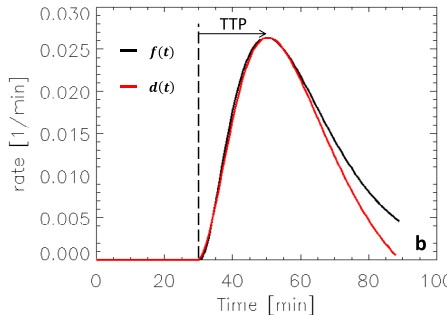

**Fig. 3 SUVr simulations: Curve fitting.** Extracellular dopamine increases, $D(t)$, simulated with Eqns [6] and [8], the corresponding changes in standardized uptake value ratio ($\Delta$SUVr) between placebo and intravenous methylphenidate, simulated with Eqns [2] and [3] for $h(t)=D(t)$, and a curve fit to the $\Delta$SUVr data using a gamma cummulative distribution (CDF), $F(t)$, given by Eqn [9] **a**. For 0.25 mg/kg methylphenidate, the time derivatives of $D(t)$ and $F(t)$, $d(t)$ and $f(t)$, had similar time-to-peak (TTP) **b**. Normal random noise (3%) was added to the SUVr time courses for placebo and IV-MP prior to compute $\Delta$SUVr in **a**. $K_1^{MP} = 0.6$ min$^{-1}$, $k_2^{MP} = 0.06$ min$^{-1}$, $k_3^{MP} = 0.5$ min$^{-1}$, and $k_4^{MP} = 0.2$ min$^{-1}$ (see ref. [31]); $K_{1r} = 0.092$ mL/min g, $k_{2r} = 0.45$ min$^{-1}$, and $R_1 = 1.154$, $k_2 = 0.45$ min$^{-1}$, and $k_{2a} = 0.065$ min$^{-1}$ (see ref. [30]); $\beta = 0.02$ min$^{-1}$.

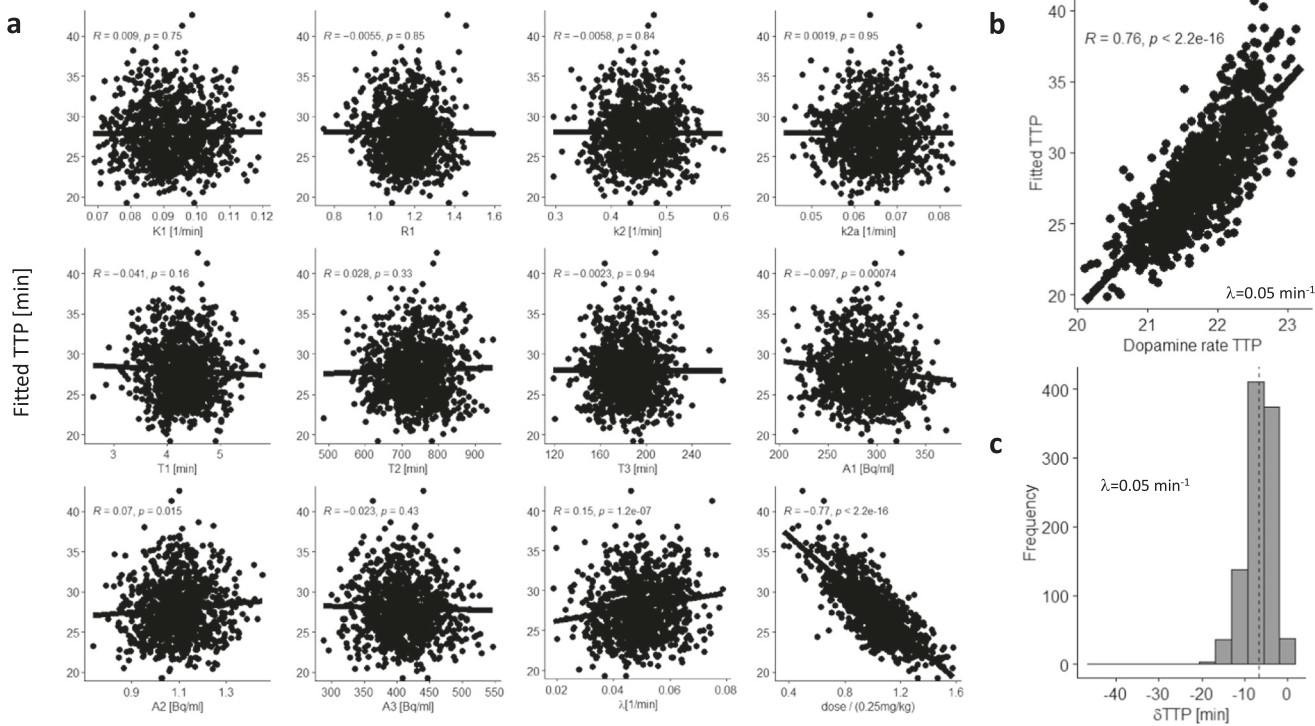

**Fig. 4 SUVr simulations for intravenous MP: Variability and accuracy in TTP:.** Scatter plots showing the lack of significant associations between time-to-peak (TTP) of the fitted gamma probability density functions, $f(t)$, and the parameters in Eqns [2], [3], and [6] ($K_1$, $R_1$, $k_2$, $k_{2a}$, $A_1$, $A_2$, $A_3$, $T_1$, $T_2$, and $T_3$; see Methods), which were randomly varied 4% within- and 10% between-simulations ($N = 1000$) with $h(t)=D(t)$; differently, fitted TTP was sensitive to 10% random variations in the dose and the decay rate of the concentration of methylphenidate (MP) in blood, $\lambda$ **a**. Fitted TTP was linearly associated across 1000 simulations with true TTP of the rate of $D(t)$ **b**. The histogram shows the skewed distribution of the TTP difference between fitted and dopamine rate TTP, dTTP **c**. Normal random noise (3%) was added to the SUVr time courses for placebo and IV-MP before computing $\Delta$SUVr. $K_1^{MP} = 0.6$ min$^{-1}$, $k_2^{MP} = 0.06$ min$^{-1}$, $k_3^{MP} = 0.5$ min$^{-1}$, and $k_4^{MP} = 0.2$ min$^{-1}$ (see ref. [31]); $K_{1r} = 0.092$ mL/min g, $k_{2r} = 0.45$ min$^{-1}$, and $R_1 = 1.154$, $k_2 = 0.45$ min$^{-1}$, and $k_{2a} = 0.065$ min$^{-1}$ (see ref. [30]); $\beta = 0.02$ min$^{-1}$.

$\Delta$SUVr showed significant DA increases in putamen as a function of time ($P < 2E{-}16$, F(1,3175) = 2775; Fig. 6b), which were higher for IV- than for oral-MP ($P = 0.04$, F(1,3175) = 4) and demonstrated a robust time-by-session interaction ($P < 2E{-}16$, F(1,3175) = 186, within-subjects ANOVA).

**Validation against a static metric of striatal DA increases.** In striatal ROIs, static SUVr values (averaged from 30 min$<t<90$ min) were strongly correlated across participants with BP$_{nd}$ assessed with the Logan plot, independently for placebo, oral- and IV-MP(R(19)>0.91; $P < 3.3E{-}08$). SUVr- and BP$_{nd}$-differences between placebo and MP also exhibited a high correlation across participants, independently for oral- and IV-MP (R(19)>0.84; $P < 3.7E{-}06$; Fig. 6a). This high correlation across subjects between the temporal average of $\Delta$SUVr and the difference in BP$_{ND}$ between PL and MP conditions, a standard measure of static DA increases, serves as additional experimental validation of $\Delta$SUVr(t) as a dynamic metric of DA increases.

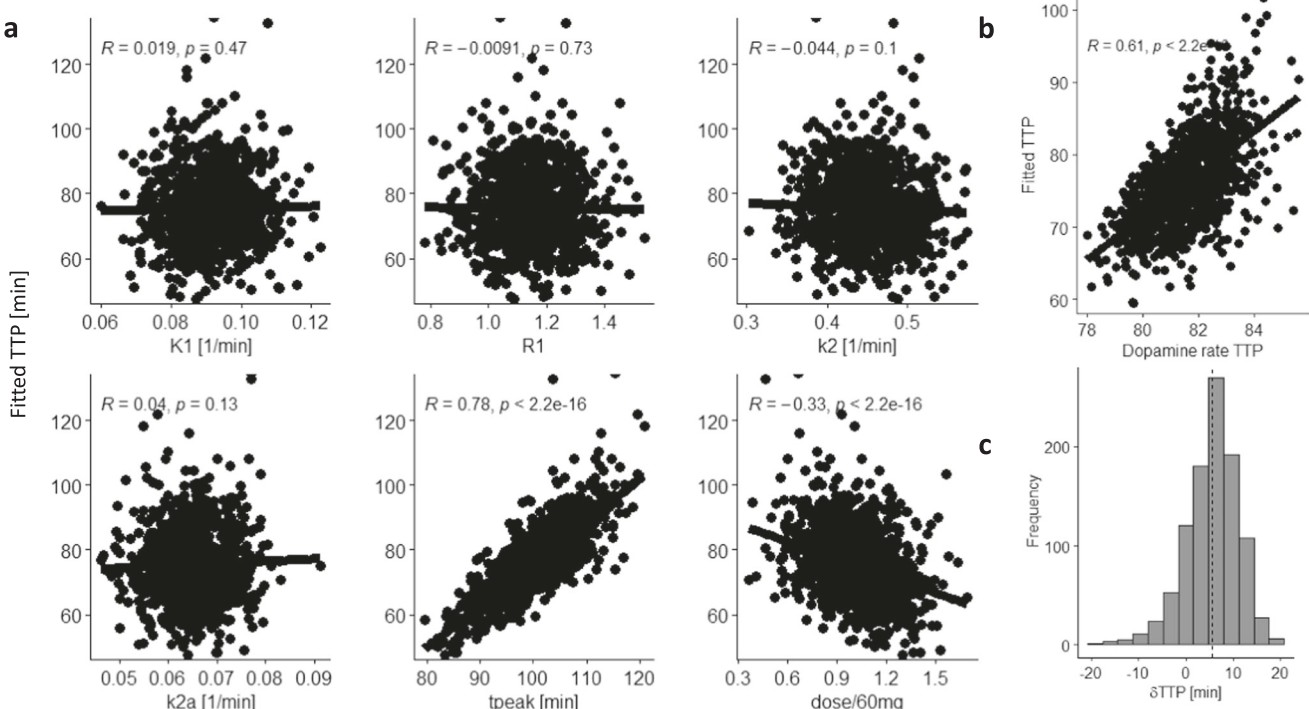

**Fig. 5 SUVr simulations for oral MP: Variability and accuracy in TTP.** Scatter plots showing the lack of significant associations between time-to-peak (TTP) of the fitted gamma probability density functions, $f(t)$, and the parameters in Eqns [2] and [6] ($K_1$, $R_1$, $k_2$, $k_{2a}$; and $t_{peak}$ see Methods), which were randomly varied 4% within- and 10% between-simulations (N = 1000) with $h(t)=D(t)$; differently, fitted TTP was sensitive to 10% random variations in $t_{peak}$ and methylphenidate (MP) dose **a**. Fitted TTP was linearly associated across 1000 simulations with true TTP of the rate of $D(t)$ **b**. The histogram shows the skewed distribution of the TTP difference between fitted and dopamine rate TTP, dTTP **c**. Normal random noise (3%) was added to the SUVr time courses for placebo and oral-MP before computing ΔSUVr. $K_1^{MP} = 0.6$ min$^{-1}$, $k_2^{MP} = 0.06$ min$^{-1}$, $k_3^{MP} = 0.5$ min$^{-1}$, and $k_4^{MP} = 0.2$ min$^{-1}$ (see ref. [31]); $K_{1r} = 0.092$ mL/min g, $k_{2r} = 0.45$ min$^{-1}$, and $R_1 = 1.154$, $k_2 = 0.45$ min$^{-1}$, and $k_{2a} = 0.065$ min$^{-1}$ (see ref. [30]); β=0.02 min$^{-1}$.

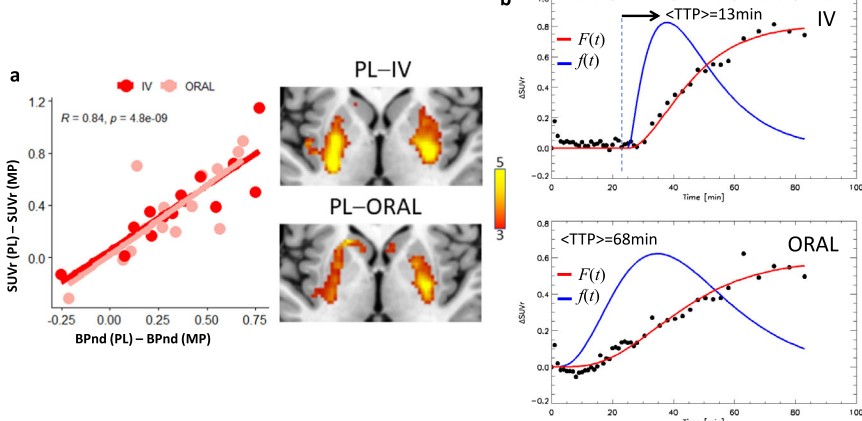

**Fig. 6 SUVr dynamics in humans. a** Differences in static standardized uptake value ratio (ΔSUVr) in the putamen (relative to the cerebellum) as a function of differences in non-displaceable binding potential (BPnd) between placebo (PL) and methylphenidate (MP) scans, for intravenous (IV) and oral sessions, and statistical t-score maps reflecting differences in BPnd between placebo and MP conditions, superimposed on axial views of the human brain at the level of the striatum. **b** Average ΔSUVr time courses (dots), and fitted gamma cumulative distribution (F) and probability (f) functions across 20 healthy adults for intravenous (IV) and oral MP. The arrow highlights the time-to-peak (TTP) of $f(t)$ since the onset of MP administration.

**Association between TTP and "high" ratings.** Peak 'high' ratings (Fig. 7a) were higher for IV- than for oral-MP (P = 0.0002; T = 4.5; df=19, paired t-test). There was large variability in TTP across individuals. For oral-MP scans, the gamma fits using Eqn [9] did not converge for 5 of the 20 participants due to poor ΔSUVr-signal to noise; in addition, fitted TTP values were flagged as outliers (>2 SD from the sample mean) for 2 (IV-MP) and 2 (oral-MP) participants. The TTP data from these participants were excluded from subsequent analyses. Fitted TTP in putamen was significantly correlated with the difference in peak 'high' ratings between MP and PL, independently for oral- and IV-MP, such that shorter TTP was associated with higher self-reports of "high" from MP (ORAL: R(13)= −0.76, IV: R(18)= −0.69; P < 0.003, two-sided; Fig. 7b). The amplitudes of fitted DA increases and its rate, $F(t)$ and $f(t)$, did not show significant correlation with differences in peak 'high' ratings between MP and PL.

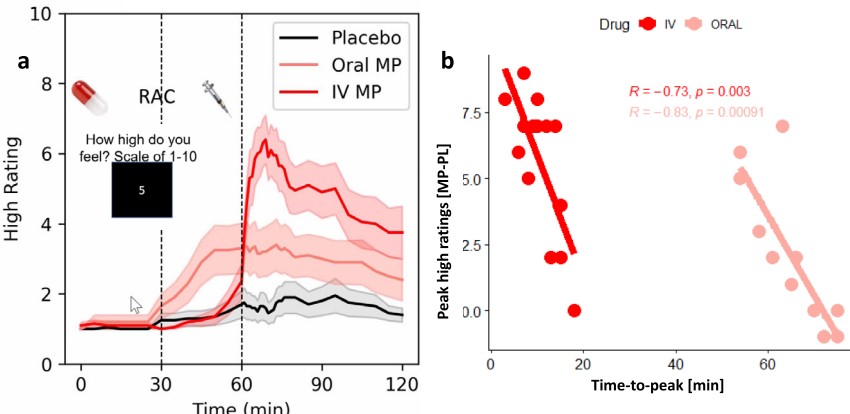

**Fig. 7 Association between shorter TTP and stronger "high" ratings. a** Average high ratings across 20 participants as a function of time during the scans. **b** Shorter time-to-peak (TTP) was associated with stronger differences in peak "high" ratings between methylphenidate (MP) and placebo (PL), independently for oral ($n = 13$) and intravenous (IV; $n = 18$) MP.

## Discussion

Dynamic PET is a unique tool to assess the rate of DA increases induced by rewarding and non-rewarding drugs, non invasively. Information about drug-related DA dynamics could be useful for a better understanding of behavioral and/or functional magnetic resonance imaging signals simultaneously collected with PET[32,33] in humans. LSSRM or ntPET have been used to model the effect of DA increases during task performance on PET signals in humans[18,21,22,29] and in preclinical studies with amphetamine[16,34] and methamphetamine[27] challenges. With simulations, we show that time-varying ΔSUVr parallels the dynamics of DA increases induced by MP in the striatum. Then, we tested the hypothesis that the intensity of the 'high' triggered by MP reflects the speed of striatal DA increases, using PET and [11C]raclopride in a study in healthy participants with a double-blind placebo-controlled within-subject design. We studied $\Delta SUVr(t)$ as a proxy for DA increases using oral (slow brain delivery) and IV-MP (fast brain delivery) challenges, in association with measured subjective responses to MP using self-reports of 'high' throughout the scan and found that DA rate TTP was associated with the perceived "high" from MP.

Brain dopamine (DA) signaling modulates movement, cognition, motivation, and reward[35,36]. Stimulant drugs that boost brain DA, such as methylphenidate (MP), are first-line therapeutics for disorders with abnormal DA signaling, including attention-deficit/hyperactivity disorder (ADHD)[37]. However, these stimulant medications are also widely misused for their rewarding effects, particularly when snorted or injected[7]. We had shown that DA increases induced by IV-MP were associated with measures of drug reward, but not those induced by oral-MP[6], and that the uptake of carbon-11 labeled MP ([11C]methylphenidate) was associated with the short-lasting duration of the "high" after IV-MP, whereas its long-lasting brain binding was not[2]. We interpreted this to indicate that it was the fast uptake and binding of MP to DAT, which we inferred would be associated with fast DA increases in the striatum, that was associated with the "high". However, the actual association between the rate of DA increases and the "high" has not been previously demonstrated.

Our simple noninvasive approach to assessing *apparent* DA increases relies on the subtraction of SUVr(t) data with and without MP. Since $BP_{ND}$ is used to quantify static DA increase[38,39] and had remarkable linear associations with static SUVr measures[40,41], we hypothesized that SUVr(t) differences between placebo and MP scans could be used to study the dynamics of DA increases elicited by MP. Consistent with our

hypothesis, the static ΔSUVr and ΔBPnd values were highly correlated across individuals. Also consistent with our hypothesis, the simulated dynamic ΔSUVr time courses were highly correlated with those of DA increases over time. Differently, the dynamics of simulated $\Delta SUVr(t)$ did not correspond well to that of DAT blockade by MP, which was characterized by $f_{occ}^{DAT}(t)$. These findings support the specificity of MP-related decreases in SUVr to the competition dynamics between [11C]raclopride and endogenous DA for binding to $D_{2/3}$ receptors. Furthermore, the amplitude of ΔSUVr increased paralleling growing DA levels elicited by increasing MP doses, thus supporting the sensitivity of ΔSUVr dynamics to time-varying changes in endogenous DA levels.

While for IV-MP no participant had a negative reaction to MP (MP-PL peak "high" rating difference ≥0), for oral MP, one participant can be said to have had a mild negative response to MP (MP-PL peak high rating difference < −1; Fig. 7b). In some people high doses of MP might trigger an aversive reaction[42] and clinical reports have described dysphoria following clinical treatment with oral MP[43]. We previously reported that disliking responses among healthy controls given a high dose of IV MP was associated with high baseline levels of striatal D2 receptors[44], suggesting that D2 receptors might modulate these responses. Indeed an hypothesized mechanism for the dysphoria, is that the initial rise of dopamine after taking MP will first bind to the inhibitory presynaptic DA autoreceptors, since they have higher affinity for dopamine than the postsynaptic DA receptors[45], leading to an initial reduction in dopamine reelase from the terminal.

With traditional models for $h(t)$ (i.e., the gamma variate function[15,30]), the simulations did not predict the $\Delta SUVr(t)$ induced by MP. However, using our mechanistic model for $h(t)$ the simulations explained the dynamics of $\Delta SUVr(t)$. Our approach with a gamma CDF function could be seen as a variation of LSSRM or ntPET methods, which do not require any specific function $h(t)$ describing the response to a pharmacological challenge. The simulations in the present study demonstrate that TTP can be estimated by fitting a gamma CDF to the experimental $\Delta SUVr(t)$ with only 2 adjustable parameters, TTP and the maximum amplitude of ΔSUVr. This model-free approach provided reliable TTP results that were associated with the 'high' elicited by MP across participants.

The fact that the dynamics of simulated $\Delta SUVr(t)$ did not correspond to DAT occupancy is not surprising since DA increases from MP are a function of 2 processes: (1) DAT

occupancy and (2) the rate of DA release by the DA terminal. Different from amphetamine, MP does not increase DA release per se, but instead its effects are due to accumulation of DA due to DAT occupancy by MP. Inasmuch as DA release is influenced by the context of administration this explains why ADHD children treated with methylphenidate show larger reductions in placebo-adjusted activity levels in the classroom than in the playground[46]. Indeed, using PET we showed that the magnitude of DA increase from MP could not be accounted by the differences in DAT blockade between individuals[4], and that the conditions in which individuals are tested influenced the magnitude of the DA increases triggered by MP[47,48].

Our model-free TTP estimations are based on gamma fits of $\Delta SUVr(t)$ measures. Prior studies that used the ntPET methodology with a gamma variate function $h(t)$ to assess the DA increases induced by ethanol found a large variability in DA TTP across participants[49]. Our simulations in the present study show the linear association between fitted and 'true' DA rate TTP. The simulations also allowed us to assess the effects of noise, protocol execution, and physiological variation from scan to scan on $\Delta SUVr(t)$ within and between individuals. The estimated errors in TTP due to reasonable injection time differences (2 min) between placebo and MP scans were within the temporal resolution (1 min) of $\Delta SUVr(t)$, suggesting that TTP is not particularly sensitive to strict protocol execution as it relates to precise timing of injections. Similarly, the simulations showed that the error in TTP due to normal variability in pharmacokinetics within and across individuals was negligible. Thus, the association between TTP and the 'high' elicited by MP suggests a biological origin for the variability in TTP in our study. Overall, our approach could help understand how alterations in TTP of dopaminergic neurotransmission could affect reward perception from drugs of abuse.

LSSRM/ntPET was not implemented in our human study. Specifically, to estimate TTP we fitted a gamma function to the $\Delta SUVr(t)$ measures. This simple approach did not require fitting $h(t)$ parameters from PET data as in LSSRM/ntPET. Thus, the experimental use of LSSRM/ntPET for measuring dopamine TTP during oral- and IV-MP challenges in humans remains to be evaluated in future studies, which could be seen as a limitation for the present study.

Model-free DA rate TTP estimations also allowed us to document that participants with faster rates of DA increases (e.g., those for whom $f(t)$ had shorter TTP) perceived the most intense 'high' during oral- and IV-MP. These findings provide strong evidence that the speed of DA increases in the striatum, which is influenced by the rate of drug uptake in the brain and is modulated by the route of drug administration, accounts for why a drug like MP can be used safely for oral ADHD treatment, whereas it can result in addiction when injected because of its reinforcing properties[7,50,51]. Thus, the faster the rate of DA increases, the more intense the "high", which would also explain why very large oral doses of stimulant drugs can also be rewarding[52].

## Methods

**Simulation of MP-related changes in SUVr**. Time-activity curves (TACs) for [11C]raclopride in the cerebellum were simulated using a one-tissue compartment model operational equation for the instantaneous tissue concentration,

$$\frac{dC_r(t)}{dt} = K_{1r}C_p(t) - k_{2r}C_r(t), \quad (1)$$

with uptake rate constant $K_{1r} = 0.092$ mL/min g and clearance rate constant $k_{2r} = 0.45$ min$^{-1}$ (see ref. [18]), and a plasmatic input function given by either the

tri-exponential function

$$C_p(t) = \begin{cases} \frac{(A_1+A_2+A_3)}{t_{peak}}t & if\ t < t_{peak} \\ \sum_{i=1}^{3}A_i\exp\left(-\frac{\ln(2)}{T_i}\left(t-t_{peak}\right)\right) & if\ t \geq t_{peak} \end{cases}, \quad (2)$$

with $\vec{A} = (A_1, A_2, A_3) = (288.6, 1.1, 409.7)Bq/ml$, $\vec{T} = (T_1, T_2, T_3) = (4.28, 735.5, 183.5)$sec, and $t_{peak} = 110$ s (see ref. [30]) for IV-MP, or the probability density function of a standard gamma distribution with time-to-peak, $t_{peak} = 90$ min, for oral-MP. Note that oral-MP is rapidly absorbed from the gastrointestinal tract achieving peak blood levels in 60 to 120 min[53]. TACs for the striatum were simulated using the LSSRM operational equation[18],

$$C_T(t) = R_1C_r(t) + k_2\int_0^t C_r(u)du - k_{2a}\int_0^t C_T(u)du - \gamma\int_0^t C_T(u)h(u)du, \quad (3)$$

with ratio of tracer delivery $R_1 = 1.154$, clearance rate constant $k_2 = 0.45$ min$^{-1}$, uptake rate constant $k_{2a} = 0.065$ min$^{-1}$, and amplitude of ligand displacement $\gamma = 0.003$, and $h(t)$ accounts for the dynamics of the dopamine–raclopride competition for $D_{2/3}$ receptor binding[30]. All parameters ($\vec{A}$, $\vec{T}$, $t_{peak}$, $K_{1r}$, $k_{2r}$, $R_1$, $k_2$, $k_{2a}$, $\gamma$, and $\lambda$) were varied 10 and 4% (100*standard deviation/mean) across 1000 simulations, using a normal random generator, to simulate between- and within-subjects physiologic variability, respectively. Two alternative mechanistic models for $h(t)$ were tested: (model 1) the fractional occupancy of DAT by MP (see below); and (model 2) the relative extracellular dopamine increases induced by MP (see below). In addition we tested the popular heuristic model for $h(t)$ which is based on a gamma variate function[15],

$$h(t) = \begin{cases} 0\ if\ t < t_D \\ \left(\frac{t-t_D}{t_p-t_D}\right)^\theta \exp\left(\theta\left[1-\frac{t-t_D}{t_p-t_D}\right]\right) & if\ t \geq t_D \end{cases} \quad (4)$$

with $t_D = 31$ min, $t_p = 45$ min, and $\theta = 15$ (see ref. [30]), but the results did not explain the dynamic changes in the experimental data. Dynamic standardized uptake value ratios were simulated as $SUVr(t) = C_T(t)/C_r(t)$, and the dynamic SUVr changes between placebo (PL) and MP conditions were simulated as $\Delta SUVr(t) = SUVr^{PL}(t) - SUVr^{MP}(t)$.

The simulations were implemented in the interactive data language (IDL, L3Harris Geospatial, Boulder, CO) and the Livermore solver for ordinary differential equations[54].

**Modeling fractional occupancy of DAT and endogenous DA increases**. We used a two-tissue compartment to assess the fractional occupancy of DAT by MP in the striatum. The time-varying concentrations of MP in the free, $C_F^{MP}$, and bound, $C_B^{MP}$, compartments were modeled using the system of ordinary differential equations:

$$\begin{cases} \frac{dC_F^{MP}(t)}{dt} = K_1^{MP}C_p^{MP}(t) - k_2^{MP}C_F^{MP}(t) - k_3^{MP}\left(1-f_{occ}^{DAT}(t)\right)C_F^{MP}(t) + k_4^{MP}C_B^{MP}(t) \\ \frac{dC_B^{MP}(t)}{dt} = k_3^{MP}\left(1-f_{occ}^{DAT}(t)\right)C_F^{MP}(t) - k_4^{MP}C_B^{MP}(t) \\ f_{occ}^{DAT}(t) = \frac{C_B^{MP}(t)}{DAT_0}, \end{cases} \quad (5)$$

where $k_i^{MP}$ are the transfer rate constants for MP, and the fractional occupancy of DAT, $f_{occ}^{DAT}(t) = C_B^{MP}(t)/DAT_0$, depends on the total concentration of dopamine transporters, $DAT_0$. These equations can be expressed in terms of the relative plasma, $R^p(t) = \frac{C_p^{MP}(t)}{DAT_0}$, and free, $R(t) = \frac{C_F^{MP}(t)}{DAT_0}$, concentrations as:

$$\begin{cases} \frac{dR(t)}{dt} = K_1^{MP}R^p(t) - k_2^{MP}R(t) - k_3^{MP}\left(1-f_{occ}^{DAT}(t)\right)R(t) + k_4^{MP}f_{occ}^{DAT}(t) \\ \frac{df_{occ}^{DAT}(t)}{dt} = k_3^{MP}\left(1-f_{occ}^{DAT}(t)\right)R(t) - k_4^{MP}f_{occ}^{DAT}(t). \end{cases} \quad (6)$$

We assumed the following transfer rate constants[31]: $K_1^{MP} = 0.6$ min$^{-1}$, $k_2^{MP} = 0.06$ min$^{-1}$, $k_3^{MP} = 0.5$ min$^{-1}$, and $k_4^{MP} = 0.2$ min$^{-1}$. For IV-MP, the simulations assumed $f_{occ}^{DAT}(t = 0) = 0$, plasma input functions, $R^p(t) = A\exp(-\lambda t)$ with $\lambda = 0.05$ min$^{-1}$, which for humans gives 13.8 min halftime for MP in the blood[55], and $A = 0.25$, consistent with DAT blockade ~70% as documented by prior studies in humans using similar IV-MP doses[6].

We used a one-tissue compartment to assess the relative concentration of endogenous DA increases, $D'(t)$, in proportion to $f_{occ}^{DAT}(t)$ (e.g., larger DAT occupancy would cause smaller DA reuptake) and the rate of clearance of extracellular DA, which would be proportional to $D'(t)$ and to the availability of DAT, $1-f_{occ}^{DAT}(t)$. Thus, the DA concentration is given by

$$\frac{dD'(t)}{dt} = \eta f_{occ}^{DAT}(t) - \beta\left[1-f_{occ}^{DAT}(t)\right]D'(t), \quad (7)$$

where $\eta$ is the rate of DA increases that reflects the firing rate of DA neurons, and $\beta$ is the dopamine clearance rate constant. Finally, $D(t) = \eta/\beta\ D'(t)$ was replaced into Eqn [7], and the scaled DA concentration, $D(t)$, was obtained by solving the

ordinary differential equation:

$$\frac{dD(t)}{dt} = \beta \left[ f_{occ}^{DAT}(t) - \left[ 1 - f_{occ}^{DAT}(t) \right] D(t) \right] \quad (8)$$

**Studies in humans**. We tested twenty healthy adults who underwent 90-min long PET scans collected in 3 randomly ordered sessions (placebo, oral-MP, and IV-MP; double-blind) while simultaneously recording their self-reported 'high' ratings (0–10) under resting conditions, using oral- and IV-MP as pharmacological challenges. In each session, each of the 20 participants was given an oral pill (60mg-MP or placebo) 30 min before injection of the PET tracer ([11C]raclopride), followed 30 min after the tracer by an IV administration (0.25 mg/kg-MP or placebo). Note that these IV- and oral-MP doses were selected because they led to roughly equivalent levels of DA transporter occupancy[6], and their administration times were chosen to ensure that peak concentrations of MP in the striatum had similar timing for oral-MP and IV-MP, relative to imaging initiation[7].

**Participants**. Twenty healthy adults (36.1 ± 9.6 years old; 9 females) were recruited to participate in the study. All individuals provided informed consent to participate in this double-blind placebo-controlled study, which was approved by the IRB at the National Institutes of Health (Combined Neurosciences White Panel; Protocol 17-AA-0178; ClinicalTrials.gov Identifier: NCT03326245). The research was performed in accordance with all relevant guidelines and regulations. Each participant was scanned on 3 different days, 40 ± 35 days apart, under different pharmacological conditions: (1) oral-MP (60 mg) and iv-placebo (3 cc saline), (2) oral-placebo and IV-MP (0.25 mg/kg in 3 cc sterile water), and (3) oral-placebo and iv-placebo. The session order was randomized across participants. Participants and researchers were blind to the nature of the drug administered orally or intravenously (MP/PL).

**PET acquisition**. The participants underwent simultaneous PET/MRI imaging in a 3 T Biograph mMR scanner (Siemens; Medical Solutions, Erlangen, Germany). All studies were initiated at noon to minimize confounds from circadian variability. Venous catheters were placed in the left dorsal hand vein for radiotracer injection, and in the right dorsal hand vein for intravenous injection of medications. Heart rate (HR), systolic and diastolic blood pressures (BPs) were continuously monitored throughout the study with an Expression MR400 patient monitor (Philips, Netherlands). Thirty minutes before tracer injection, either 60 mg of MP or placebo was administered p.o. The participant was then positioned in the scanner. Earplugs were used to minimize scanner noise and padding was used to minimize head motion. A T1 weighted dual-echo image was collected for attenuation correction using an ultrashort-TE (UTE) sequence (192[3] matrix, 1.56 mm isotropic resolution, TR = 11.94 ms, TE = 0.07 and 2.46 ms), and T1-weighted 3D magnetization-prepared gradient-echo (MPRAGE; TR/TI/TE = 2200/1000/4.25 ms; FA = 9°, 1 mm isotropic resolution) was used to map brain structure. List mode PET emission data were acquired continuously for 90 min and initiated immediately after a manual bolus injection of [11C]raclopride (dose = 15.7 ± 1.9 mCi; duration 5–10 s). Thirty minutes after tracer injection, either 0.25 mg/kg MP or placebo were manually injected i.v. as a ~30-s bolus. The participants were instructed to remain as still as possible and to relax and keep their eyes open during scanning.

**High ratings**. Self-reports of "High" rating prompts were displayed on a projector using a program (E-Prime Version 3.0) designed to minimize visual stimulation. A white cross was presented at central fixation on a black screen. Participants were instructed to stay awake, relaxed, to look at the cross, and not think of anything in particular. Occasionally, the cross would turn into a number for 10 s, and participants responded with a rating to the question: "How high do you feel right now, on a scale of 1–10, with 1 being minimum and 10 being maximum?". The first number presented at the start of each scanning session was always 1, and subsequent presentations matched the participant's high rating from the prior time point. Participants used a button box in their right hand to record responses. A button pressed with the right middle finger moved the rating up, one digit at a time, whereas the other button pressed with the right index finger moved the scale down. High rating prompts occurred every 5 min from the onset of oral MP administration; then, at the onset of IV-MP administration, prompts occurred every minute for 20 min—this faster sampling was chosen to capture the fast changes in reward during the first 20 min after the onset of IV-MP administration[6,56,57]; then, prompts occurred every 5 min again until the end of scanning. High rating maxima were normally distributed (p > 0.5, Shapiro-Wilk's normality test), did not have outliers across study day or Drug condition, and were higher for IV-MP than for oral-MP, and for oral-MP than for placebo, regardless of study day. Within-subjects ANOVA demonstrated a strong effect of MP (P < 1E-07) but no effect of study day (p = 0.4) on high rating maxima, suggesting minimal carryover effects.

**MRI preprocessing**. The minimal preprocessing pipelines of the Human Connectome Project (HCP)[58] were used for image processing. Specifically, FreeSurfer 5.3.0 (http://surfer.nmr.mgh.harvard.edu) was used for automatic segmentation of anatomical MRI scans into cortical and subcortical gray matter ROIs[59], and the

FSL Software Library (version 5.0; http://www.fmrib.ox.ac.uk/fsl)[60] was used for spatial normalization to MNI space.

**PET image reconstruction**. A 3-dimensional ordered-subset expectation-maximization (OSEM) algorithm[61] with 3 iterations, 21 subsets, an all-pass filter, 344 × 344 × 127 matrix, and a model of the point spread function of the system was used for PET image reconstruction. The reconstructed PET time series consisted of 48 time windows (30 frames of 1 min, followed by 12 frames of 2.5 min, and 6 frames of 5 min) each with 2.086-mm in-plane resolution and 2.032-mm slice thickness. Attenuation coefficients (μ-maps) estimated from the UTE data using a fully convolutional neural network[62] were used to correct for scattering and attenuation of the head, the MRI table, the gantry, and the radiofrequency coil. Standardized uptake values (SUVs) for [11C]raclopride were calculated after normalization for body weight and injected dose and spatially normalized to MNI space using HCP pipelines. Relative SUV time series, SUVr(t), were computed in MNI space by normalizing each SUV volume by its mean SUV in the cerebellum, as defined in individual FreeSurfer segmentations.

**PET image analysis**. Time-activity curves were computed for putamen, caudate, and ventral striatum and cerebellum from SUV time series using individual FreeSurfer segmentations. The Logan Plot graphical analysis for reversible systems using the cerebellum as the reference tissue and equilibration time t* = 20 min was used to map the distribution volume ratio (DVR) and non-displaceable binding potential (BPnd)[63], independently for each participant and session.

**Non-linear fitting**. The amplitude, A, and the shape, s, of the gamma cumulative distribution function

$$F(t) = \frac{A}{\Gamma(s)} \int_0^t e^{-x} x^{s-1} dx, \quad (9)$$

were adjusted to fit F(t) to the ΔSUVr(t) data in IDL using the Levenberg-Marquardt algorithm for non-linear least-squares fitting[64]. The time-to-peak of the corresponding probability density function, f(t) = dF(t)/dt, was calculated as TTP = s − 1. For oral-MP scans, the fits did not converge for 5 of the 20 participants due to poor signal-to-noise.

**Statistics and reproducibility**. Within-participants analysis of variance (ANOVA) in R was used to assess the main effects of time and session, as well as time-by-session interactions on DA increases and peak "high" ratings. We used within-participants ANOVA in the statistical parametric mapping package (SPM12; Wellcome Trust Centre for Neuroimaging, London, UK) to assess the statistical significance of BPnd in the brain. The voxelwise inference was based on a familywise error (FWE) correction for multiple comparisons[65]. Specifically, voxels were considered statistically significant if they had $P_{FWE} < 0.05$, corrected for multiple comparisons with the random field theory using a cluster defining threshold p < 0.001.

**Reporting summary**. Further information on research design is available in the Nature Portfolio Reporting Summary linked to this article.

## Data availability
The data used for the figures has been uploaded to Figshare (https://figshare.com/) and is now accessible (https://doi.org/10.6084/m9.figshare.21948209). The source data used during the current study will be available from the corresponding author on reasonable request.

## Code availability
The IDL code used in the current study will be available from the corresponding author on reasonable request.

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

## Acknowledgements

This work was accomplished with support from the National Institute on Alcohol Abuse and Alcoholism (ZIAAA000550).

## Author contributions

D.T. and N.D.V. designed the study and interpreted the data; D.T. developed code and performed simulations and statistical analysis; J.L. critiqued and revised the computational methods; P.M., M.V.Y, D.K., D.F., K.M., C.B., E.D., A.J, W.T.W., and G.J.W. collected data; E.S.K. provided image preprocessing; K.Y. developed software; J.B. provided access to imaging resources. D.T. and N.D.V wrote the manuscript with contributions from all co-authors.

## Funding

## Competing interests

The authors declare no competing interests.
