## [Peer Review File · Communications Biology]

Reviewers' comments:

Reviewer #1 (Remarks to the Author):

The manuscript by Tomasi and colleagues examines whether speed of dopamine time-to-peak (i.e., rate of increase) in the striatum underlies self-reports of drug "high" in response to oral vs intravenous administration of methylphenidate (MP). This is the first work in humans that I am aware of that relates rates of dopamine increases to drug reward / high. The novel modeling of PET imaging in humans (and simulations) reveals critical information on the role of dopamine and drug kinetics and important work not only for understanding abuse liability of MP, but also how kinetics relate to abuse liability of all psychostimulants. The manuscript is timely, clear and well written. The experimental design is sound, and statistics are appropriate. The author conclusion are justified. There are a few relatively minor comments to follow up on.

1. The authors seem to use interchangeably the self-report "high" (as the outcome variable) with the psychological construct/term of reward. Are humans truly reporting reward in this context, or could high mean a subjective assessment of some other sensation? Discussion is needed to justify use of the use of the term reward.
2. Negative peak high ratings in Figure 6B are interesting but do not receive much discussion. Could negative ratings be considered aversion? A significant number of patients have negative ratings (MP – Placebo) in the oral condition. Is this aversion to MP or a placebo high?
3. Was there any sense of carryover effects from self report measures across conditions. Perhaps conditions were counterbalanced in their order. Could this be related to point #2 if a "high" is compared against possible expectations set in earlier exposures.
4. Dopamine release was used throughout the manuscript to describe elevations in striatal dopamine, although authors do distinguish between uptake inhibition/DAT blockade and exogenous release mechanisms. Perhaps release in the general sense could be switch to dopamine elevations (or similar) when relevant to separate the concept of exogenous release mechanisms.

Reviewer #2 (Remarks to the Author):

The authors utilize a unique and innovative method to examine dopamine increases in the striatum in individuals following oral and IV methylphenidate. A novel finding includes the demonstration that drug reward is associated with faster increases in DA induced by methylphenidate. I found a few typos and grammatical errors, but otherwise, the manuscript was well-written and the experimental design, including statistics, were appropriate. I have only a few minor comments. Abstract: Last sentence- methylphenidate does not cause "release" but rather blocks reuptake. Therefore, it would be more appropriate to replace "release" with "increase". I find it interesting that the dynamics of simulated $\Delta\text{SUVr}(t)$ did not correspond to DAT occupancy, but rather just DA increases. However, I am confused by the authors' discussion of this finding on page 13. It would be helpful to expand this section.

Reviewer #3 (Remarks to the Author):

The manuscript, 'Time-varying SUVr reflects the dynamics of dopamine increases during methylphenidate challenges in humans' by Tomasi and colleagues proposes dynamic estimates of dopamine kinetics by simple subtraction of the SUV from two scans and fitting the difference of these curves to a gamma c.d.f. The work presents simulations to support the proposed method, but these simulations lack a full examination of the experimental conditions in the later presented human data.

Since the advantages of the proposed method over established dynamic analysis methods are not obvious from this paper, the impact of this work is low. More specific comments are as follows:

A central claim of this paper seems to be that the dynamic SUVr model with a gamma CDF to describe radiotracer dynamics fits better than LSRRM (or ntPET) models. A major strength of these models is the flexibility to estimate both the timing and magnitude of dopamine release, but the presented implementation appears to hold those parameters fixed based on literature values, which undermines the *raison d'être* for those models. Relatedly, it is curious that LSRRM is used to simulate data but LSRRM (or ntPET) is not used to analyze the human data. It is unclear what advantages the proposed approach provides over these established methods.

A key result of this paper seems to be the use of a newer, simplified analysis approach to quantify timing and speed of oral vs. i.v. methylphenidate. However, the presented studies feature differences in timing of these challenges (oral 30 min before radiotracer injection, i.v. 30 min after radiotracer injection). This design choice, which is reasonable to best capture peak effects of each challenge, results in methylphenidate likely affecting rates of radiotracer brain entry (e.g., K_1) for the oral challenge but not the I.v. challenge. This possible effect on the proposed analysis is not convincingly characterized in the simulations, since the simulations only examine robustness with 1% variability in parameters, which substantially underestimates likely effects of drug challenge on these parameters. Simulations supporting the performance of TTP quantification of data from oral methylphenidate 30 min prior to radiotracer injection are similarly missing.

In human data there is report of significant variability in TTP across individuals. It seems that the simulations performed could help address whether this variability represents biological variability or lack of precision in the TTP analysis.

In the analysis of human data it is not clear how the various d.o.f. values for the F-tests/ANOVA were calculated.

"The speed of DA increases in striatum, estimated for $30 < t < 50$ min using linear regression analysis" – this approach was not validated, and strictly speaking does not only represent the reduction in radiotracer binding from the challenge, but rather that function convolved with the clearance of radiotracer from the tissue (i.e., k_2). Importantly to this design, since oral MP was administered 30 min prior to injection this comparison is occurring at different times relative to drug administration, which seems to be an apples to oranges comparison.

Figure 5A was difficult to understand. What units are the color scale in?

Response to Reviewer #1

Comment: 'The manuscript by Tomasi and colleagues examines whether speed of dopamine time-to-peak (i.e., rate of increase) in the striatum underlies self-reports of drug "high" in response to oral vs intravenous administration of methylphenidate (MP). This is the first work in humans that I am aware of that relates rates of dopamine increases to drug reward / high. The novel modeling of PET imaging in humans (and simulations) reveals critical information on the role of dopamine and drug kinetics and important work not only for understanding abuse liability of MP, but also how kinetics relate to abuse liability of all psychostimulants. The manuscript is timely, clear and well written. The experimental design is sound, and statistics are appropriate. The author conclusion are justified. There are a few relatively minor comments to follow up on.'

Response: We are thankful for the supporting comment.

Comment 1: 'The authors seem to use interchangeably the self-report "high" (as the outcome variable) with the psychological construct/term of reward. Are humans truly reporting reward in this context, or could high mean a subjective assessment of some other sensation? Discussion is needed to justify use of the use of the term reward.'

Response: We now refer to 'high ratings' instead of 'reward'.

Comment 2: 'Negative peak high ratings in Figure 6B are interesting but do not receive much discussion. Could negative ratings be considered aversion? A significant number of patients have negative ratings (MP – Placebo) in the oral condition. Is this aversion to MP or a placebo high?'

Response: We thank the reviewer for bringing this to our attention. While for IV-MP no participant had a negative reaction to MP (MP-PL peak high rating difference ≥ 0), for oral MP, one participant can be said to have had a very mild negative rating to MP (MP-PL peak high rating difference < -1 ; Fig 6B). In some people high doses of MP might trigger an aversive reaction¹ and clinical reports have described dysphoria following clinical treatment with oral MP². We have also reported disliking of IV-MP challenges in participants with high baseline levels of D2R³. It has been proposed that a potential mechanism for the dysphoria reported in some individuals with oral MP treatment is that when MP enters slowly into the brain the initial rise of dopamine will first bind to the inhibitory presynaptic DA autoreceptors, since they have higher affinity for dopamine than the postsynaptic DA receptors⁴ leading to a an initial net reduction in dopamine signaling. We added this paragraph to Discussion.

Comment 3. 'Was there any sense of carryover effects from self-report measures across conditions. Perhaps conditions were counterbalanced in their order. Could this be related to point #2 if a "high" is compared against possible expectations set in earlier exposures.'

Response: We thank the reviewer for highlighting this important point. Our study had a double-blind within-subject placebo-controlled design in which each participant was scanned on 3 different study days, 40±35 days apart, in which session order (PL, IV-MP, and oral-MP) was randomized across participants to further minimize carryover effects. High maxima were normally distributed ($p>0.5$, Shapiro-Wilk's normality test) and did not have outliers across study day or Drug condition. As shown in the Figure, the high maxima were higher for IV-MP than for Oral-MP than for PL, regardless of study day. Within-subjects ANOVA demonstrated a strong effect of Drug ($P<1E-07$) but no effect of study day ($p=0.4$), suggesting minimal carryover effects. We added this information to Methods.

Comment 4: 'Dopamine release was used throughout the manuscript to describe elevations in striatal dopamine, although authors do distinguish between uptake inhibition/DAT blockade and exogenous release mechanisms. Perhaps release in the general sense could be switch to dopamine elevations (or similar) when relevant to separate the concept of exogenous release mechanisms.'

Response: We now refer to 'DA increases' instead of 'DA release'.

Response to Reviewer #2

Comment: 'The authors utilize a unique and innovative method to examine dopamine increases in the striatum in individuals following oral and IV methylphenidate. A novel finding includes the demonstration that drug reward is associated with faster increases in DA induced by methylphenidate. I found a few typos and grammatical errors, but otherwise, the manuscript was well-written and the experimental design, including statistics, were appropriate. I have only a few minor comments.'

Response: We are thankful for the supporting comment.

Comment 1: 'Abstract: Last sentence- methylphenidate does not cause "release" but rather blocks reuptake. Therefore, it would be more appropriate to replace "release" with "increase".'

Response: We now refer to 'DA increases' instead of 'DA release'.

Comment 2: 'I find it interesting that the dynamics of simulated $\Delta\text{SUVr}(t)$ did not correspond to DAT occupancy, but rather just DA increases. However, I am confused by the authors' discussion of this finding on page 13. It would be helpful to expand this section.'

Response: We apologize for the lack of clarity in this regard. The fact that the dynamics of simulated $\Delta\text{SUVr}(t)$ did not correspond to DAT occupancy is not surprising since DA increases from MP are a function of 2 processes: 1) DAT occupancy and 2) the rate of DA release by the DA terminal. Different from amphetamine MP does not increase DA release per se but instead its effects are due to accumulation of DA due to DAT occupancy by MP, which blocks its removal from the synapse. Inasmuch as DA release is influenced by the context of administration this explains why ADHD children treated with methylphenidate show larger reductions in placebo-adjusted activity levels in the classroom than in the playground⁵. Indeed using PET we showed that the magnitude of DA increase from MP could not be accounted by the differences in DAT blockade between individuals⁶, and that the conditions in which

individuals are tested influenced the magnitude of DA increases triggered by MP^{7,8}. We have expanded the Discussion as suggested by the reviewer.

Response to Reviewer #3

Comment 1: 'Since the advantages of the proposed method over established dynamic analysis methods are not obvious from this paper, the impact of this work is low.'

Response: We respectfully disagree with the reviewer. The proposed method provides a new way to estimate time-to-peak, which is valuable to measure the speed of DA increases elicited by psychostimulant drugs non-invasively in humans. Reviewer 1 stated "This is the first work in humans that I am aware of that relates rates of dopamine increases to drug reward / high. The novel modeling of PET imaging in humans (and simulations) reveals critical information on the role of dopamine and drug kinetics and important work not only for understanding abuse liability of MP, but also how kinetics relate to abuse liability of all psychostimulants."

Comment 2: 'A central claim of this paper seems to be that the dynamic SUVr model with a gamma CDF to describe radiotracer dynamics fits better than LSRRM (or ntPET) models. A major strength of these models is the flexibility to estimate both the timing and magnitude of dopamine release, but the presented implementation appears to hold those parameters fixed based on literature values, which undermines the raison d'être for those models. Relatedly, it is curious that LSRRM is used to simulate data but LSRRM (or ntPET) is not used to analyze the human data. It is unclear what advantages the proposed approach provides over these established methods.'

Response: The purpose of the simulations was to show that $\Delta\text{SUVr}(t)$ reflects the dynamics of DA increases elicited by MP. Specifically, the simulations demonstrated that, with traditional models for $h(t)$ (i.e., the gamma variate function^{9,10}), LSRRM did not predict well $\Delta\text{SUVr}(t)$ induced by MP. However, using our mechanistic model for $h(t)$ LSRRM explained the dynamics of $\Delta\text{SUVr}(t)$. One limitation of the LSRRM (or ntPET) is the large number of freely adjustable parameters (R_1 , k_2 , k_{2a} , and γ) in the model, which can be hard to fit in human studies. We did not aim to quantify these parameters but to measure the time-to-peak (TTP) of the rate of DA increases estimated from $\Delta\text{SUVr}(t)$. The simulations demonstrate that TTP can be estimated by fitting a gamma CDF to the experimental $\Delta\text{SUVr}(t)$ with only 2 adjustable parameters, TTP and the maximum amplitude of ΔSUVr . This model-free approach provided reliable TTP results that were associated with the 'high' elicited by MP across participants. We added this clarification to Discussion.

Comment 3: 'A key result of this paper seems to be the use of a newer, simplified analysis approach to quantify timing and speed of oral vs. i.v. methylphenidate. However, the presented studies feature differences in timing of these challenges (oral 30 min before radiotracer injection, i.v. 30 min after radiotracer injection). This design choice, which is reasonable to best capture peak effects of each challenge, results in methylphenidate likely affecting rates of radiotracer brain entry (e.g., K_1) for the oral challenge but not the i.v. challenge. This possible effect on the proposed analysis is not convincingly characterized in the simulations, since the simulations only examine robustness with 1% variability in

parameters, which substantially underestimates likely effects of drug challenge on these parameters. Simulations supporting the performance of TTP quantification of data from oral methylphenidate 30 min prior to radiotracer injection are similarly missing.'

Response: We thank the reviewer for raising this issue. To assess potential R1-differences between IV- and oral MP we reanalyzed the experimental data using the SRTM (see Figure below). We found that MP decreased R1 in gray matter compared to placebo (PL>IV-MP: CI= [0.072, 0.002], $p < 0.04$, $df=18$; PL>oral-MP: CI= [0.050, 0.001], $p < 0.03$, $df=19$; 2-sided paired t-test), which is consistent with moderate vasoconstriction effects of MP. However, there were no significant difference in R1 between IV- and oral-MP conditions ($p > 0.47$; $df=18$, 2-sided paired t-test) such that this analysis did not support vasoconstriction differences between IV- and oral-MP.

In our simulations “all parameters (\vec{A} , \vec{T} , t_{peak} , K_{1r} , k_{2r} , R_1 , k_2 , k_{2a} , g , and l) were varied 5% and 2% (100*standard deviation/mean)” (page 15), not 1% which was erroneously mentioned in page 7. To address the reviewer’s concern, the revised Fig 4 now presents simulations with larger within- (4%) and between-subjects (10%) variability in the parameters of the model. In addition, the revised manuscript includes simulations for oral-MP, which as displayed in an additional figure (Fig 4b), as requested by the reviewer. The simulations in Fig 4aA and 4bA also show that TTP is not sensitive to R1 variability.

Comment 4: ‘In human data there is report of significant variability in TTP across individuals. It seems that the simulations performed could help address whether this variability represents biological variability or lack of precision in the TTP analysis.’

Response: The reviewer is correct. Prior studies that used the ntPET methodology with a gamma variate function $h(t)$ to assess the DA increases induced by ethanol found a large variability in DA TTP across participants¹¹. In the present study our simulations show the linear association between fitted and ‘true’ DA rate TTP elicited by MP. The simulations also allowed us to assess the effects of noise, protocol execution, and physiological variation from scan to scan on $\Delta SUVr(t)$ within and between individuals. The estimated errors in TTP due to reasonable injection time differences (2 min) between placebo and MP scans were within the temporal resolution (1 min) of $\Delta SUVr(t)$, suggesting that TTP is not particularly sensitive to strict protocol execution as it relates to precise timing of injections. Thus, the association between TTP and the ‘high’ elicited by MP suggests a biological origin for the variability in TTP in our study. Overall, our approach could help understand how alterations in TTP of dopaminergic neurotransmission could affect reward perception from drugs of abuse. We added this to Discussion.

Comment 5: ‘In the analysis of human data it is not clear how the various d.o.f. values for the F-tests/ANOVA were calculated.’

Response: We thank the reviewer for alerting us of the errors in the reported d.o.f. For within-subjects ANOVA the revised version documents d.o.f. calculated using the R function `avov`.

Comment 6: ‘ “The speed of DA increases in striatum, estimated for 30<t<50 min using linear regression analysis” – this approach was not validated, and strictly speaking does not only represent the reduction in radiotracer binding from the challenge, but rather that function convolved with the clearance of radiotracer from the tissue (i.e., k_2). Importantly to this design, since oral MP was administered 30 min prior to injection this comparison is occurring at different times relative to drug administration, which seems to be an apples to oranges comparison.’

Response: We removed the controversial sentence.

Comment 7: ‘Figure 5A was difficult to understand. What units are the color scale in?’

Response: We apologize for the missing information. Fig 5A shows a t-score color map. This information has been added to the legend.

REFERENCES

1. Lucas, P., Gardner, D., Wolkowitz, O. & Cowdry, R. Dysphoria associated with methylphenidate infusion in borderline personality disorder. *Am J Psychiatry* **144**, 1577-1579 (1987).
2. Naguy, A. Duloxetine Alleviates Stimulant Dysphoria, Helps With Enuresis, and Complements Cognitive Response in an Adolescent With Attention-Deficit/Hyperactivity Disorder. *Prim Care Companion CNS Disord* **18**, doi: 10.4088/PCC.4016l01957. PMID: 28033457. (2016).
3. Volkow, N., *et al.* Prediction of reinforcing responses to psychostimulants in humans by brain dopamine D2 receptor levels. *Am J Psychiatry* **156**, 1440-1443 (1999).
4. Ford, C. The role of D2-autoreceptors in regulating dopamine neuron activity and transmission. *Neuroscience* **282**, 13-22 (2014).
5. Swanson, J., *et al.* Efficacy of a new pattern of delivery of methylphenidate for the treatment of ADHD: effects on activity level in the classroom and on the playground. *Journal of the American Academy of Child and Adolescent Psychiatry* **41**, 1306-1314 (2002).
6. Volkow, N., *et al.* Relationship between blockade of dopamine transporters by oral methylphenidate and the increases in extracellular dopamine: Therapeutic implications. *Synapse* **43**, 181-187 (2002).
7. Volkow, N., *et al.* "Nonhedonic" food motivation in humans involves dopamine in the dorsal striatum and methylphenidate amplifies this effect. *Synapse* **44**, 175-180 (2002).
8. Volkow, N., *et al.* Evidence that methylphenidate enhances the saliency of a mathematical task by increasing dopamine in the human brain. *Am J Psychiatry* **161**, 1173-1180 (2004).
9. Itrace, Z., *et al.* Bayesian Estimation of the ntPET Model in Single-Scan Competition PET Studies. *Front Physiol* **11**, 498 (2020).
10. Normandin, M., Schiffer, W. & Morris, E. A linear model for estimation of neurotransmitter response profiles from dynamic PET data. *Neuroimage* **59**, 2689-2699 (2012).
11. Constantinescu, C., *et al.* Estimation from PET data of transient changes in dopamine concentration induced by alcohol: support for a non-parametric signal estimation method. *Phys Med Biol* **53**, 1353-1367 (2008).

Reviewers' comments:

Reviewer #1 (Remarks to the Author):

The authors have done a good job responding to comments and revising their manuscript. I have no further comments.

Reviewer #2 (Remarks to the Author):

Thank you for your thorough responses to my questions as well as those from the other reviewers. I have no further issues with the manuscript and look forward to its publication.

Reviewer #3 (Remarks to the Author):

This manuscript revision addresses a number of errors/typos from the initial draft.

I do not dispute that the findings of timing of peak response with MP 'high' is interesting or novel, in fact I very much agree and think it is important this finding is shared with the field.

There seems to remain some confusion on using LSSRM. In both the paper and reviewer responses the claim is made that 'LSSRM did not predict well $\Delta\text{SUVR}(t)$ '. But from my read of the paper $h(t)$ parameters were fixed a priori, which means that LSSRM was not implemented. The claim 'LSSRM did not predict well $\Delta\text{SUVR}(t)$ ' cannot be made unless the $h(t)$ parameters are flexibly estimated. Similarly, if the claim that 'the large number of freely adjustable parameters (in ntPET) ... can be hard to fit in human studies' is to be made, there should be statistical support from the presented data for this claim or it should be removed.

Ultimately, the presented analysis method is not really so different from LSSRM/ntPET as it is a variation of these methods. LSSRM or ntPET do not require any specific function describing the response from a challenge. Decaying exponent or gamma variate functions were used because they were appropriate in the context of prior work. In this sense, the present use of a gamma CDF function($F(t)$) is really just a different form of the response function in the operational lp-ntPET equation ($h(t)$), and their characterized timing of that function(e.g. TTP) likely would be even more robust and show similar relationships with behavioral measures.

The lack of analyses that would allow for a direct conversation with the extant literature using these dynamic analysis methods remains a major limitation.

Responses to Reviewers

Reviewer #1:

Comment: 'The authors have done a good job responding to comments and revising their manuscript. I have no further comments.'

Response: Thanks for your support.

Reviewer #2:

Comment: 'Thank you for your thorough responses to my questions as well as those from the other reviewers. I have no further issues with the manuscript and look forward to its publication.'

Response: Thanks for your support.

Reviewer #3:

Comment 1: 'There seems to remain some confusion on using LSSRM. In both the paper and reviewer responses the claim is made that 'LSSRM did not predict well $\Delta\text{SUVr}(t)$ '. But from my read of the paper $h(t)$ parameters were fixed a priori, which means that LSSRM was not implemented. The claim 'LSSRM did not predict well $\Delta\text{SUVr}(t)$ ' cannot be made unless the $h(t)$ parameters are flexibly estimated. Similarly, if the claim that 'the large number of freely adjustable parameters (in ntPET) ... can be hard to fit in human studies' is to be made, there should be statistical support from the presented data for this claim or it should be removed.'

Response: We agree with the reviewer. The unsupported claim in Discussion was removed as requested by the reviewer.

Comment 2: 'Ultimately, the presented analysis method is not really so different from LSSRM/ntPET as it is a variation of these methods. LSSRM or ntPET do not require any specific function describing the response from a challenge. Decaying exponent or gamma variate functions were used because they were appropriate in the context of prior work. In this sense, the present use of a gamma CDF function ($F(t)$) is really just a different form of the response function in the operational lp-ntPET equation ($h(t)$), and their characterized timing of that function (e.g., TTP) likely would be even more robust and show similar relationships with behavioral measures.'

Response: We agree with the reviewer on that our approach is similar to LSSRM or ntPET, which do not require any specific function describing the response from a challenge. The paragraph was rewarded to emphasize that 'our approach with a gamma CDF function could be seen as a variation of LSSRM or ntPET methods, which do not require any specific function $h(t)$ describing the response to a pharmacological challenge'. Our study did not aim to implement LSSRM for the estimation of $h(t)$ parameters but to show that ΔSUVr can be used as a proxy for dopamine increases. The limitations paragraph now highlights that the experimental use of LSSRM/ntPET for measuring dopamine TTP

during oral- and IV-MP challenges in humans remains to be evaluated in future studies.

Comment 3: 'The lack of analyses that would allow for a direct conversation with the extant literature using these dynamic analysis methods remains a major limitation'

Response: We agree with the reviewer. The limitations paragraph now highlights that "LSSRM/ntPET was not implemented in our human study. Specifically, to estimate TTP we fitted a gamma function to the $\Delta\text{SUVR}(t)$ measures. This simple approach did not require fitting $h(t)$ parameters from PET data as in LSSRM/ntPET. Thus, the experimental use of LSSRM/ntPET for measuring dopamine TTP during oral- and IV-MP challenges in humans remains to be evaluated in future studies, which could be seen as a limitation for the present study.

REVIEWERS' COMMENTS:

Reviewer #3 (Remarks to the Author):

The manuscript now presents a more fair discussion of prior work, although it would have been very interesting to see results from directly comparing different dynamic analysis methods with these data.

Responses to Reviewers

Reviewer #3:

Comment: 'The manuscript now presents a more fair discussion of prior work, although it would have been very interesting to see results from directly comparing different dynamic analysis methods with these data.'

Response: While it's an interesting recommendation it will be very lengthy for us to develop the methodology for LSSRM parametric fitting. Also, if the comparison between the LSSRM approach and our model-free approach reveals differences, it could be difficult to judge which method is more accurate. In our opinion, the comparison of these different dynamic analysis methods deserves a comprehensive analysis and full paper.